# SiMFy: A Simple Yet Effective Approach for Temporal Knowledge Graph Reasoning

**Zhengtao Liu, Lei Tan, Mengfan Li, Yao Wan, Hai Jin, Xuanhua Shi**[†]
National Engineering Research Center for Big Data Technology and System,
Services Computing Technology and System Lab, Cluster and Grid Computing Lab,
School of Computer Science and Technology,
Huazhong University of Science and Technology, Wuhan, China
{zhengtaoliu,tanlei,limf,wanyao,hjin,xhshi}@hust.edu.cn

## Abstract

*Temporal Knowledge Graph (TKG)* reasoning, which focuses on leveraging temporal information to infer future facts in knowledge graphs, plays a vital role in knowledge graph completion. Typically, existing works for this task design graph neural networks and recurrent neural networks to respectively capture the structural and temporal information in KGs. Despite their effectiveness, in our practice, we find that they tend to suffer the issues of *low training efficiency* and *insufficient generalization ability*, which can be attributed to the over design of model architectures. To this end, this paper aims to figure out whether the current complex model architectures are necessary for temporal knowledge graph reasoning. As a result, we put forward a simple yet effective approach (termed SiMFy), which simply utilizes *multilayer perceptron* (MLP) to model the structural dependencies of events and adopts a fixed-frequency strategy to incorporate historical frequency during inference. Extensive experiments on real-world datasets demonstrate that our SiMFy can reach state-of-the-art performance with the following strengths: 1) faster convergence speed and better generalization ability; 2) a much smaller time consumption in the training process; and 3) better ability to capture the structural dependencies of events in KGs. These results provide evidence that the substitution of complex models with simpler counterparts is a feasible strategy.[1]

## 1 Introduction

*Knowledge Graphs (KGs)*, which represent events as triples $(s, r, o)$, facilitate a wide range of natural language processing tasks, including semantic search (Bonatti et al., 2019; Wang et al., 2020), product recommendations (Xie et al., 2021), and question-answering systems (Saxena et al., 2020).

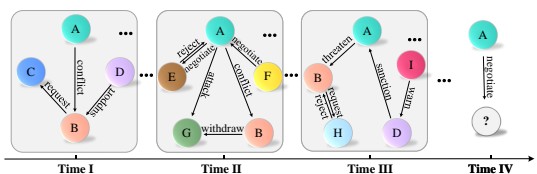

Figure 1: An illustration of TKG reasoning, showing knowledge graphs at three distinct timestamps. Directed edges represent relations, originating from the subject entity and terminating at the object entity. Our objective is to predict the object entity at the final timestamp.

However, traditional KGs struggle to effectively handle facts that have temporal characteristics. Therefore, *Temporal Knowledge Graphs (TKGs)* have been introduced to tackle this challenge (Wang et al., 2023), which incorporate a time dimension $t$ and store facts as quadruples $(s, r, o, t)$, e.g., (*Germany*, *negotiate*, *Russia*, *2022-05*).

However, in real-world scenarios, TKGs are often incomplete, highlighting the vital importance of TKG reasoning, which aims to predict future facts by utilizing the temporal information. As illustrated in Figure 1, given the TKGs associated with Entity A at the timestamps of *Time I*, *Time II*, and *Time III*, our objective is to predict a quadruple containing an unknown entity, i.e., (*A*, *negotiate*, *?*, *Time IV*). To achieve this goal, we can observe a significant structural dependency of facts along the timeline. That is, interactions between Entity A and Entity B at *Time I* could influence the interaction patterns at *Time II*, which, in turn, might provide clues for *Time III*. We also notice that historical events may recur. For instance, Entity A and Entity B had conflicts at both *Time I* and *Time II* timestamps. These observations provide us valuable insights into TKG reasoning.

In the TKG reasoning task, events to be predicted can typically be classified into two main types: historical events and unseen events (Han et al., 2021). Historical events, also known as

---

[1]The source code and datasets are available at https://github.com/CGCL-codes/SiMFy.

[†]Xuanhua Shi is the corresponding author.

repetitive pattern, refer to facts that have already occurred in the historical KG sequence. Unseen events refer to events that have not occurred in the historical KG sequence. Many methods (Jin et al., 2020; Zhu et al., 2021) can predict historical events effectively by modeling the historical KG sequence auto-regressively. However, for unseen events, it is necessary to consider both the information of structural dependency and temporality of entities and relations. Typically, existing works (Liu et al., 2022; Li et al., 2022) apply *Graph Neural Networks (GNNs)* and *Recurrent Neural Networks (RNNs)* as encoders to capture the structural and temporal information in the historical KG sequence. Then, a translational model, such as Conv-TransE (Shang et al., 2019), is used as a decoder to obtain predicted entities. Despite their effectiveness, these approaches are conceptually and technically complex due to their advanced model architectures. Moreover, these complex methods tend to suffer the issues of *low training efficiency* and *insufficient generalization ability*.

In this paper, we aim to answer the following research question: *Are these complex model architectures indispensable for temporal knowledge graph reasoning?* As a solution, we design a **Simple MLP-Frequency-based model** (SiMFy) to evaluate it against other complex baselines. Specifically, we use a one-layer MLP to jointly model entities and relations, capturing structural and temporal information in TKGs. This allows us to obtain embedding vectors for entities and relations, which are then used to calculate the similarity between queries and candidate entities, resulting in preliminary candidate entity scores. Next, we calculate the historical frequency scores of candidate entities based on their historical KG sequences. Finally, in the inference stage, the two scores mentioned above are combined using a coefficient $\alpha$ to obtain the final entity scores. Through extensive experiments comparing it with existing state-of-the-art models, we find that our model can achieve comparable performance while having higher training efficiency and better generalization ability.

Furthermore, we conduct a series of empirical studies to investigate the performance of SiMFy and existing complex models under specific conditions. We are the first to analyze the performance of MLP and GNNs in capturing unseen entities. Our findings indicate that MLP performs comparably to GNN in capturing structural dependency information for the TKG reasoning task. Furthermore, we investigate the question of whether historical frequency information should be incorporated into the model training process, which has not been explored before. Unlike the most current mainstream methods which directly incorporate historical frequency information of entities during training, SiMFy utilizes conceptually simple features to model the repetitive pattern of TKGs, which is fixed in training. Through empirical experiments, we validate the effectiveness of this fixed-frequency strategy adopted by SiMFy.

The contributions of this paper are as follows.
- We design a simple MLP-based-only model, called SiMFy, which achieves state-of-the-art performance on four widely-used datasets (i.e., ICEWS14, ICEWS18, ICEWS05-15, and GDELT), demonstrating the effectiveness of the simple model architecture.
- We have performed extensive experiments to analyze the convergence speed, generalization ability, and training consumption of SiMFy and existing complex models. The empirical evidence demonstrates that a simple model architecture like SiMFy enjoys faster convergence, better generalization ability, and higher efficiency.
- The performance of SiMFy could motivate future research to rethink the significance of the simpler model architecture and the potential value of MLP-based models in TKGs.

## 2   Related Work

**Static KG Reasoning** In static knowledge graphs, inference is done to deduce unknown facts from known ones. Existing approaches for inference can be categorized into embedding-based, tensor decomposition-based, and neural network-based. Embedding-based methods, represented by the classic method TransE (Bordes et al., 2013), consider relations as translational transformations. Later models like KG2E (He et al., 2015) optimize the handling of diverse relations and improve model scalability. Tensor decomposition methods represent the graph as a tensor which is then decomposed. Models like RESCAL (Nickel et al., 2011) use this to capture interactions between entities and relations. As for the neural network models like ConvE (Dettmers et al., 2018) and KG-BART (Liu et al., 2021), ConvE enhances inference capabilities through deep feature learning and leveraging graph

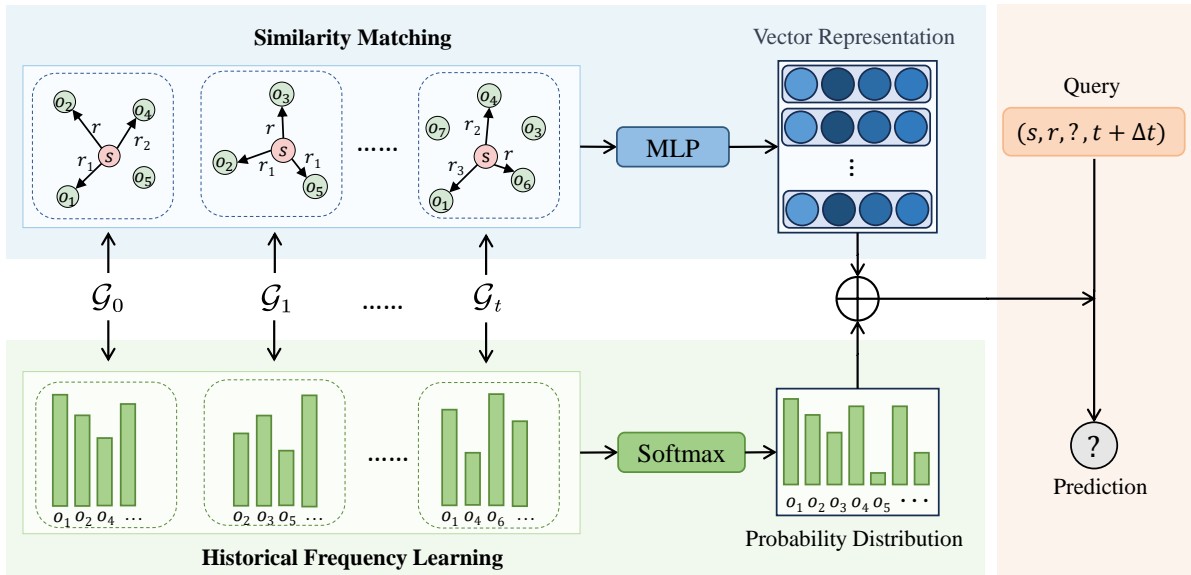

Figure 2: Overall model architecture of SiMFy. Given a query $(s, r, ?, t + \Delta t)$, the Similarity Matching module learns the similarity score between the query and each candidate entity $o$, and the Historical Frequency Learning module learns the historical frequency score between them, then the two scores are combined to generate the final probability distribution of entities.

structures, and KG-BART utilizes pretrained language models augmented with knowledge graphs to not only enhance inference but also improve the generation of commonsense-reasoned text.

**Temporal KG Reasoning**   Temporal knowledge graph inference, which takes into account the temporal evolution of events, generally falls into two categories: interpolation-based and extrapolation-based inference.   Interpolation-based inference aims to guess unknown facts within a known time range.  TTransE (Jiang et al., 2016) integrates temporal information into the TransE (Bordes et al., 2013) model using recursive neural networks, while HyTE (Dasgupta et al., 2018) designs a unique hyperplane to embed time into the entity-relation space. TeMP (Hu et al., 2022) addresses time sparsity and variability issues by combining neural message passing and temporal dynamic methods.

On the other hand, extrapolation-based inference, which forecasts unknown future facts, is garnering increasing attention.  RE-NET (Jin et al., 2020) employs an encoder and aggregator to model past facts, while HIP (He et al., 2021) utilizes temporal, structural, and repetitive information. xERTE (Han et al., 2021) offers a novel framework for predicting future facts and CyGNet (Zhu et al., 2021) introduces a creative copy-mechanism used in natural language generation tasks before. Reinforce-

ment learning is adopted by CluSTeR (Li et al., 2021a), which infers answers from induced clues, and TimeTraveler (Sun et al., 2021), which uses historical knowledge graph snapshots for answer search. CEN (Li et al., 2022) adopts a length-aware convolutional network to model the KG sequence dynamically and GHT (Sun et al., 2022) is the first method to introduce a transformer into the TKG reasoning task. Finally, DA-Net (Liu et al., 2022) and CENET (Xu et al., 2023) propose unique event prediction models. DA-Net learns distributed attention to future events, while CENET distinguishes likely entities for a given query using a historical contrastive learning framework.

## 3   Problem Formulation

### 3.1   Temporal Knowledge Graph

A *Temporal Knowledge Graph* (TKG) $\mathcal{G}$ is a sequence of KGs $(\mathcal{G}_0, \mathcal{G}_1, \ldots, \mathcal{G}_t)$ arranged in order of their timestamp $t$. $\mathcal{G} = \{\mathcal{E}, \mathcal{R}\}$, where $\mathcal{E}$ stands for the set of entities, and $\mathcal{R}$ for the set of relations. Each $\mathcal{G}_t = \{\mathcal{E}_t, \mathcal{R}_t\}$, where $\mathcal{E}_t \subseteq \mathcal{E}$ and $\mathcal{R}_t \subseteq \mathcal{R}$ are the sets of entities and relations at timestamp $t$ respectively. In $\mathcal{G}_t$, facts are represented as quadruples $(s, r, o, t)$, where $s, o \in \mathcal{E}_t$ and $r \in \mathcal{R}_t$.

### 3.2   TKG Reasoning

TKG reasoning seeks to forecast either the subject entity $s$ or the object entity $o$ based on the historical

KG sequence $\{\mathcal{G}_0, \mathcal{G}_1, \ldots, \mathcal{G}_t\}$. When presented with a query of the form $(?, r, o, t + \Delta t)$, the task is to identify the subject entity $s$, while for a query like $(s, r, ?, t + \Delta t)$, the aim is to predict the object entity $o$. We use $\mathbf{E} \in \mathbb{R}^{|\mathcal{E}| \times d}$ and $\mathbf{R} \in \mathbb{R}^{|\mathcal{R}| \times d}$ to express the embeddings of all entities and all relations respectively. Boldfaced $\mathbf{s}, \mathbf{r}, \mathbf{o}$ are used to denote the embedding vectors of $s$, $r$, and $o$ with a dimension of $d$. In our work, we specifically concentrate on the task of object entity prediction.

# 4 Our Approach

Here we elaborate our proposed *Simple MLP-Frequency-based model* (SiMFy) for temporal graph reasoning. As illustrated in Figure 2, SiMFy mainly involves two modules: the Similarity Matching module and the Historical Frequency Learning module. These two modules generate corresponding scores for candidate entities. Afterward, a weight-based inference process is utilized to determine the final result of the reasoning. In the subsequent sections, we will provide a comprehensive introduction to our proposed method.

## 4.1 Similarity Matching

Given a query $q = (s, r, ?, t + \Delta t)$, the Similarity Matching module is implemented by one-layer-MLP to calculate the similarity between $q$ and each candidate entity $o$ to obtain the matching scores. Specifically, it generates a latent context vector $\mathbf{H}_{s,r} \in \mathbb{R}^{|\mathcal{E}|}$ for query $q$, which scores the similarity of different object entities with the query:

$$\mathbf{H}_{s,r} = \tanh\left(\mathbf{W}\left[\mathbf{s}, \mathbf{r}\right] + \mathbf{b}\right)\mathbf{E}^T \quad (1)$$

where $\mathbf{s} \in \mathbf{E}, \mathbf{r} \in \mathbf{R}$, and $[\mathbf{s}, \mathbf{r}]$ denotes the concatenation of $\mathbf{s}$ and $\mathbf{r}$. We use one-layer-MLP to aggregate the query's information. Here, $\mathbf{W} \in \mathbb{R}^{d \times 2d}$ and $\mathbf{b} \in \mathbb{R}^d$ are trainable parameters. $\tanh$ is the activation function of the layer, then the layer's output is multiplied by $\mathbf{E}$ to obtain the final similarity vector, where each element represents the similarity score between the corresponding entity $o \in \mathcal{E}$ and the query $q$. The learning objective of the similarity matching is to minimize the NCE loss (Gutmann and Hyvärinen, 2010) $\mathcal{L}$, as described below:

$$\mathcal{L} = -\sum_q \log \frac{\exp\left(\mathbf{H}_{s,r}\left(o_i\right)\right)}{\sum_{o_j \in \mathcal{E}} \exp\left(\mathbf{H}_{s,r}\left(o_j\right)\right)} \quad (2)$$

where $o_i$ is the ground truth object entity corresponding to the given query $q$. Finally, we obtain the following similarity score by utilizing the soft-max function:

$$S_{sim}^{(s,r)} = \mathrm{softmax}(\mathbf{H}_{s,r}) \quad (3)$$

## 4.2 Historical Frequency Learning

Given a query $q = (s, r, ?, t + \Delta t)$ and the historical KG sequence $\{\mathcal{G}_0, \mathcal{G}_1, \ldots, \mathcal{G}_t\}$, the Historical Frequency Learning module aims to obtain the historical frequency scores. Specifically, for every timestamp $t' \leq t$, we first investigate the frequencies of historical entities $f_{t'}^{(s,r)} \in \mathbb{R}^{|\mathcal{E}|}$, as follows:

$$f_{t'}^{(s,r)}(o) = \sum_{x \in \mathcal{G}_{t'}} \mathbb{I}[x = (s, r, o, t')] \quad (4)$$

where $\mathbb{I}[\cdot]$ is an indicator function, yielding 1 if $[\cdot]$ is true and 0 otherwise. Then we add up the frequency information of all timestamps $t' \leq t$ to obtain the historical frequency information of the query as follows:

$$\mathbf{F}_{t+\Delta t}^{(s,r)} = f_0^{(s,r)} + f_1^{(s,r)} + \cdots + f_t^{(s,r)} \quad (5)$$

where $\mathbf{F}_{t+\Delta t}^{(s,r)} \in \mathbb{R}^{|\mathcal{E}|}$ is an $|\mathcal{E}|$-dimensional vector where each element represents the corresponding historical frequency of the candidate object entities. Finally, we obtain the following historical frequency score by utilizing the softmax function:

$$S_{freq}^{(s,r)} = \mathrm{softmax}(k \cdot \mathbf{F}_{t+\Delta t}^{(s,r)}) \quad (6)$$

where $k$ is a hyperparameter to balance the extremely small values.

## 4.3 Inference

In the inference stage, a coefficient $\alpha \in [0, 1]$ is integrated to balance the weight between the similarity score $S_{sim}^{(s,r)}$ and the historical frequency score $S_{freq}^{(s,r)}$. These two scores are merged to determine the final probability distribution of candidate entities, then the model chooses the object with the highest probability as the final prediction, as defined below:

$$\mathbf{P}(o|s, r, t + \Delta t) = \alpha \cdot S_{sim}^{(s,r)}(o) + (1 - \alpha) \cdot S_{freq}^{(s,r)}(o)$$

$$o_{t+\Delta t} = \arg\max_{o \in \mathcal{E}} \mathbf{P}(o|s, r, t + \Delta t) \quad (7)$$

where $\mathbf{P}(o|s, r, t + \Delta t)$ is an $|\mathcal{E}|$-dimensional vector which stores the final probability of all entities.

Table 1: Experimental results of the entity prediction task (**raw metrics**) on ICEWS14, ICEWS18, ICEWS05-15, and GDELT datasets. The best results are boldfaced and the ones of second-best are underlined.

| Model | ICEWS14 (raw) | | | | ICEWS18 (raw) | | | | ICEWS05-15 (raw) | | | | GDELT (raw) | | | |
|---|---|---|---|---|---|---|---|---|---|---|---|---|---|---|---|---|
| | MRR | Hits@1 | Hits@3 | Hits@10 | MRR | Hits@1 | Hits@3 | Hits@10 | MRR | Hits@1 | Hits@3 | Hits@10 | MRR | Hits@1 | Hits@3 | Hits@10 |
| RotatE | 25.71 | 16.41 | 29.01 | 45.16 | 14.53 | 6.47 | 15.78 | 31.86 | 19.01 | 10.42 | 21.35 | 36.92 | 3.62 | 0.52 | 2.26 | 8.37 |
| ConvE | 30.30 | 21.30 | 34.42 | 47.89 | 22.81 | 13.63 | 25.83 | 41.43 | 31.40 | 21.56 | 35.70 | 50.96 | 18.37 | 11.29 | 19.36 | 32.13 |
| Conv-TransE | 31.50 | 22.46 | 34.98 | 50.03 | 23.22 | 14.26 | 26.13 | 41.34 | 30.28 | 20.79 | 33.80 | 49.95 | 19.07 | 11.85 | 20.32 | 33.14 |
| R-GCN | 28.03 | 19.42 | 31.95 | 44.83 | 15.05 | 8.13 | 16.49 | 29.00 | 27.13 | 18.83 | 30.41 | 43.16 | 12.17 | 7.40 | 12.37 | 20.63 |
| TTransE | 12.86 | 3.14 | 15.72 | 33.65 | 8.44 | 1.85 | 8.95 | 22.38 | 16.53 | 5.51 | 20.77 | 39.26 | 5.53 | 0.46 | 4.97 | 15.37 |
| HyTE | 16.78 | 2.13 | 24.84 | 43.94 | 7.41 | 3.10 | 7.33 | 16.01 | 16.05 | 6.53 | 20.20 | 34.72 | 6.69 | 0.01 | 7.57 | 19.06 |
| TA-DistMult | 26.22 | 16.83 | 29.72 | 45.23 | 16.42 | 8.60 | 18.13 | 32.51 | 27.51 | 17.57 | 31.46 | 47.32 | 10.34 | 4.44 | 10.44 | 21.63 |
| xERTE | 32.23 | 24.29 | 36.41 | 48.76 | 27.98 | **19.26** | 32.43 | 46.00 | 38.07 | 28.45 | 43.92 | 57.62 | - | - | - | - |
| RE-NET | 35.77 | 25.99 | 40.10 | 54.87 | 26.17 | 16.43 | 29.89 | 44.37 | 36.86 | 26.24 | 41.85 | 57.60 | 19.60 | 12.03 | 20.56 | 33.89 |
| CyGNet | 35.06 | 25.78 | 39.00 | 53.42 | 24.80 | 15.37 | 28.29 | 43.46 | 36.24 | 25.65 | 41.54 | 56.26 | 17.99 | 11.10 | 19.03 | 31.26 |
| CENET | 36.36 | 27.32 | 40.02 | 54.39 | 26.43 | 17.57 | 29.44 | 44.12 | 38.74 | 28.57 | 43.39 | 58.37 | - | - | - | - |
| GHT | 37.40 | 27.77 | 41.66 | 56.19 | 27.40 | 18.08 | 30.76 | 45.76 | 41.50 | 30.79 | 46.85 | **62.73** | 20.04 | 12.68 | 21.37 | 34.42 |
| CEN | 36.32 | 26.83 | 40.20 | 54.93 | 26.70 | 17.34 | 30.33 | 44.84 | 36.66 | 26.27 | 41.04 | 57.12 | 19.27 | 11.96 | 20.60 | 33.50 |
| RE-GCN | 37.58 | 27.49 | 41.92 | 57.69 | 27.93 | 18.18 | 31.63 | 47.07 | 37.79 | 26.86 | 42.91 | 58.93 | 19.07 | 12.16 | 20.20 | 32.39 |
| **SiMFy** | **39.54** | **29.56** | **44.56** | **59.18** | **28.65** | 18.99 | **32.45** | **47.62** | **41.83** | **31.30** | **47.02** | 62.11 | **21.33** | **13.38** | **23.16** | **36.87** |

## 5 Experiments

### 5.1 Experimental Setup

**Dataset** During the evaluation, we adopt four publicly available TKG datasets: ICEWS14 (Li et al., 2021b), ICEWS05-15 (García-Durán et al., 2018), ICEWS18 (Jin et al., 2020), and GDELT (Leetaru and Schrodt, 2013). The first three datasets which originate from the *Integrated Crisis Early Warning System* (Boschee et al., 2015) (ICEWS) comprise a diverse range of political facts accompanied by time annotations, such as (*European Union*, *Praise or endorse*, *Kosovo*, *2018/09/29*). *Global Database of Events, Language, and Tone* (GDELT) is a much larger dataset that records data every 15 minutes. Following the dataset split strategy proposed by (Jin et al., 2020), we divide these datasets into training, validation, and test sets, adhering to an 80%, 10%, and 10% proportion by timestamps.

**Evaluation Metrics** We evaluate our approach to the task of entity prediction, where the objective is to predict the absent object entity for a given entity-relation pair, assessing whether the ground truth entity ranks higher than other entities. We present the results in terms of Hits@1/3/10 and *Mean Reciprocal Rank* (MRR).

During the evaluation stage, two settings are commonly employed: filtered setting and raw setting. In terms of the filtered setting, for each query, we treat all triples absent from the training, validation, and test sets to be negative samples. This implies that, when computing rankings, we disregard all triples known to be true.

In contrast, the raw setting does not involve fil-tering any triples, meaning that all possible triples are considered when calculating rankings. In the context of TKG reasoning task, each quadruple $(s, r, o, t)$ is unique due to the inclusion of time-stamp information. This suggests that the filtered setting holds little practical relevance when evaluating model performance. Therefore, we adopt the raw setting for our experiments.

**Baselines** We assess how well our SiMFy model performs in comparison to three other types of previously proposed models: (1) Static reasoning methods, including RotatE (Sun et al., 2019), Conv-TransE (Shang et al., 2019), ConvE (Dettmers et al., 2018), and R-GCN (Schlichtkrull et al., 2018). (2) Dynamic interpolated reasoning methods, including TTransE (Jiang et al., 2016), HyTE (Das-gupta et al., 2018), and TA-DistMult (García-Durán et al., 2018). (3) Dynamic extrapolated reasoning methods, including xERTE (Han et al., 2021), RE-NET (Jin et al., 2020), CyGNet (Zhu et al., 2021), CENET (Xu et al., 2023), GHT (Sun et al., 2022), CEN (Li et al., 2022) and RE-GCN (Li et al., 2021b).

### 5.2 Implementation Details

For the baselines of CyGNet, CENET, CEN, and RE-GCN, we rerun these models using their open-source code on four datasets with their default parameter settings. Since the code of GHT is not open-source, we provide the findings from their publication. It should be noted that we utilize the offline version of CEN to ensure fairness. Some results of static and dynamic interpolated reasoning approaches are adopted from (Li et al., 2021b).

We implement our SiMFy model using PyTorch.

Table 2: Ablation study of SiMFy on ICEWS14

| Model | MRR | Hits@1 | Hits@3 | Hits@10 |
|---|---|---|---|---|
| **SiMFy** | **39.54** | **29.56** | **44.56** | **59.18** |
| SiMFy w.o. HF | 35.97 | 26.03 | 40.85 | 55.39 |
| SiMFy w.o. SM | 31.18 | 24.07 | 34.51 | 44.34 |

The dimension of entity and relation embeddings is set to 200. We use Adam (Kingma and Ba, 2015) as the optimizer, with a learning rate of 0.001 and a weight decay of 0.00001. The hyperparameter $k$ is set to 2 and the $\alpha$ is set to 0.001. The batch size is set to 1024 and the training epoch is limited to 30. All experiments were conducted on a Tesla V100.

## 5.3 Model Results

Table 1 shows the experimental results of SiMFy compared with other baselines on four datasets. It is clear that SiMFy performs better than other baselines in most cases. For static reasoning methods, their results are very poor because they do not consider temporal information. Dynamic interpolated reasoning methods like TTransE only encode time information without considering the evolution of temporal KG sequences, so they cannot achieve good results. It is worth noting that for GCN-based models CEN and RE-GCN, as well as the models most pertinent to our model, RE-NET, CyGNet, and CENET, SiMFy outperforms them. This demonstrates the effectiveness of simple models based on MLPs and shows that MLPs can to some extent replace GCNs. Please refer to Section 6.3 for a more in-depth analysis.

## 5.4 Ablation Study

We perform ablation experiments on the ICEWS14 dataset to better understand the impact of each SiMFy module, and the results are displayed in Table 2. SiMFy w.o. HF refers to SiMFy without the Historical Frequency Learning module, while SiMFy w.o. SM represents SiMFy without the Similarity Matching module. It can be observed that SiMFy w.o. HF performs better than SiMFy w.o. SM. This is because the Similarity Matching module captures many unseen events in the TKGs, which also demonstrates the ability of MLP in understanding the structural dependencies of events. Therefore, SiMFy w.o. SM, which only considers historical repetitive patterns, results in a significant **drop of 21.1% and 25.1%** in terms of MRR and Hits@10 respectively compared to SiMFy. Despite

having some capability to handle historical events, SiMFy w.o. HF also experiences a **drop of 9%** in performance, which further confirms the effectiveness of the Historical Frequency Learning module.

## 5.5 Case Study

To further illustrate 1) SiMFy's ability to predict unseen entities by capturing the structural dependency information of KGs, and 2) SiMFy's ability to predict repetitive events using historical frequency, we present two cases in the ICEWS18 dataset.

- In the first case, the query is (*Dharamvira Gandhi*, *Criticize or denounce*, *?*, *t*), and the correct objective entity is *Government (India)*. However, upon investigating the historical KG sequence of this query, we find that the triple (*Dharamvira Gandhi*, *Criticize or denounce*, *Government (India)*) has not occurred before. This indicates that there is no repetitive pattern for the quadruple *(Dharamvira Gandhi, Criticize or denounce, Government (India), t)*, making it an unseen event with *Government (India)* as the corresponding unseen entity. Nevertheless, due to SiMFy's extraction of structural dependency information from the historical KG sequence, we find that the candidate entity *Government (India)* appears as the top first in the ranking list generated by the model. This indicates that SiMFy has successfully captured the correct unseen entity, despite its absence in the historical KGs.

- In the second case, the query is (*Zdravko Maric*, *Make statement*, *?*, *t*), and the correct objective entity is *Government (Croatia)*. By analyzing the historical KG sequence of this query, we find that the triple (*Zdravko Maric*, *Make statement*, *Government (Croatia)*) has the highest frequency of occurrence, significantly surpassing other cases. SiMFy, utilizing the learned historical frequency information, recognizes this repetitive pattern and memorizes it. As a result, in the final ranking list generated by the model, the correct answer *Government (Croatia)* occupies the top position. This aligns with our real-world understanding that Zdravko Maric served as the Minister of Finance in the Croatian government, making statements on behalf of the government frequently.

## 6 More Empirical Results and Analysis

Here, we show more empirical results and analysis to reveal the strengths of our proposed SiMFy.

Table 3: The comparison results of the MLP and GCN-based models' ability to predict unseen events

| Model | ICEWS14 unseen | | | ICEWS18 unseen | | | ICEWS05-15 unseen | | |
|---|---|---|---|---|---|---|---|---|---|
| | MRR | Hits@3 | Hits@10 | MRR | Hits@3 | Hits@10 | MRR | Hits@3 | Hits@10 |
| RGCN+ConvTransE | 13.90 | 14.92 | 26.60 | 11.15 | 11.54 | 21.90 | 12.56 | 13.34 | 25.03 |
| RGCN+MLP | 14.10 | 15.03 | 26.90 | 10.70 | 11.03 | 20.86 | 13.21 | 13.95 | 25.86 |
| **MLP** | **15.53** | **17.01** | **30.97** | **11.28** | **11.59** | **22.91** | **14.00** | **15.06** | **27.65** |

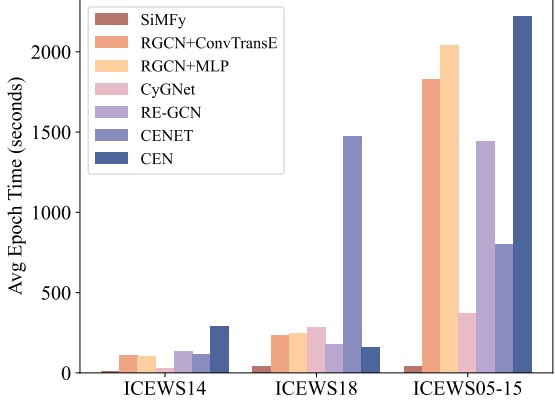

Figure 3: Time consumption of different models

## 6.1 Convergence Speed and Generalization

To gain a deeper understanding of the model's performance, we will closely examine the dynamics of both training MRR and evaluation MRR. Following (Cong et al., 2023), we will also discuss the important generalization gap (the absolute difference between the MRR scores obtained during training and evaluation) of different models. In addition to SiMFy, we also use four existing complex baseline models, including CyGNet, RE-GCN, CENET, and CEN, which are most relevant to our work, as experimental comparisons. Moreover, based on RGCN, we also reconstruct two new GCN-based models as our comparative baselines. The first one is RGCN+ConvTransE, where ConvTransE (Shang et al., 2019) is used as the decoder. The second one is RGCN+MLP, using MLP as the decoder. Figure 4 illustrates the efficacy of the models on the most representative ICEWS05-15 dataset, from which the following conclusions can be drawn:

• As the slope change of the training MRR curve can reflect the convergence speed of the model (i.e., a slope close to 0 indicates model convergence), we can see that our model always converges after several epochs, indicating a very fast convergence speed.

• After each epoch, we save separate models and evaluate their performance on the test set. Figure 4b illustrates that our model not only achieves the highest MRR but also maintains stability and smoothness in the curve after several epochs. In contrast, the MRR curves of other baselines show varying degrees of oscillation.

• The generalization ability of a model refers to its performance on unseen data, that is, the adaptability of the model to new data. Therefore, the smaller the generalization gap, the better the model's generalization ability. As shown in Figure 4c, our model has the smallest generalization gap, showing its strong generalization ability.

• Combining these three figures, we can also observe an interesting phenomenon. Models based on GCN, represented by CEN (Li et al., 2022), even if their performance on the test set fails to enhance after numerous epochs, their MRR on the training set keeps rising. This means that these GCN-based models have overfitting problems, which is also reflected in the increasing generalization gap.

## 6.2 Low Time Consumption

Compared to other complex models, SiMFy has much lower training time consumption because the model parameter numbers of SiMFy are much smaller. This could be attributed to the concise and effective model structure of SiMFy. Taking the typical GCN-based method as an example, our SiMFy is simply composed of an MLP layer that maps the input embedding to the output embedding. The GCN-based models also perform the multi-step propagation operation, which will significantly increase the model parameters, especially when the depth of the GCN is increased.

For the ICEWS14, ICEWS18, and ICEWS05-15 benchmarks, we calculate the average training time consumption of our model SiMFy and other baselines over 30 epochs. As shown in Figure 3, the time cost is highest for the ICEWS05-15 dataset, followed by the ICEWS18 dataset, and the lowest for the ICEWS14 dataset. However, on all three

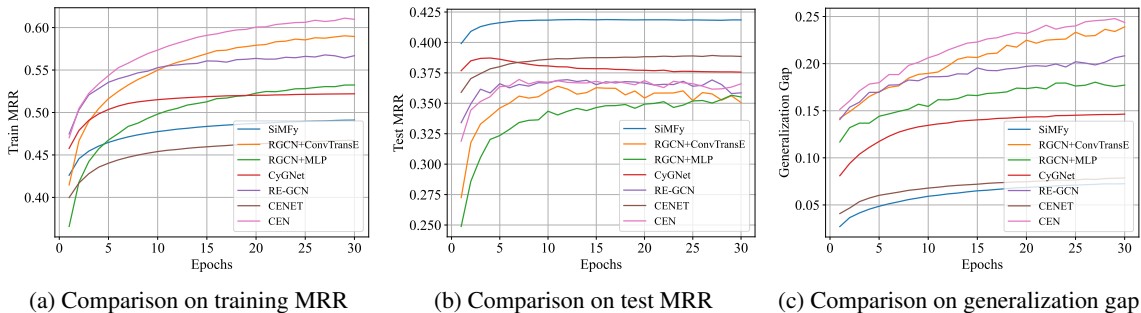

(a) Comparison on training MRR  (b) Comparison on test MRR  (c) Comparison on generalization gap

Figure 4: Comparison on the training MRR, test MRR, and generalization gap for the 30 training epochs on ICEWS05-15. Results on other ICEWS datasets can be found in Appendix A.4.

datasets, SiMFy exhibits the lowest time consumption compared to other baselines during the training process, making it more efficient and greatly improving resource utilization. Furthermore, we can observe that the training time consumption of SiMFy does not exhibit significant fluctuations across different datasets.

### 6.3 Ability to Capture Unseen Events

To predict the unseen events that do not appear in the historical KG sequence, a natural idea is using the historical evolutionary information of other events that interact with them, which is also known as the event structure dependency information. The mainstream view is that GCN can better capture this structural dependence, interestingly, we find that MLP and GCN have similar abilities to capture unseen events. Therefore, given the high training cost of GCN, we doubt whether it is necessary to apply it to TKG reasoning task. We collect all unseen events from the test set for the three ICEWS datasets, as indicated in Table 3, and then we evaluate the MLP module and the GCN, respectively. The results indicate that the performance of MLP is even slightly superior to that of GCN, which also validates the effectiveness of our MLP-based model.

Table 4: Comparative experiments of frequency on ICEWS14

| Model | MRR | Hits@1 | Hits@3 | Hits@10 |
|---|---|---|---|---|
| **MLP-F.F.** | **39.54** | **29.56** | **44.56** | **59.18** |
| MLP-ONLY | 35.97 | 26.03 | 40.85 | 55.39 |
| MLP-COPY | 35.96 | 26.54 | 40.20 | 54.02 |

### 6.4 Fixed Frequency in Training

SiMFy utilizes the historical frequency information of events only during the inference stage

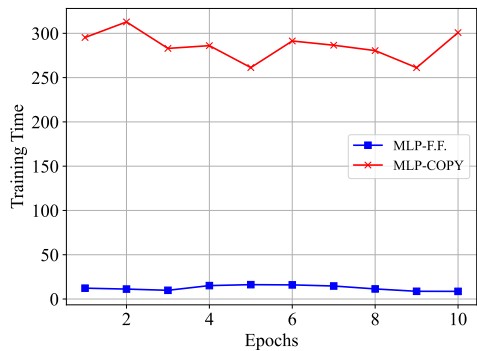

Figure 5: Training time in first 10 epochs

(fixed-frequency in training). However, many existing state-of-the-art methods, like CENET and CyGNet, combine event embeddings with historical frequency during the training process and iteratively update the model parameters. To validate which approach is more beneficial for the TKG reasoning task, we conduct comparative experiments on the ICEWS14 dataset by adopting the copy-mechanism-based learning strategy used in CyGNet and CENET. The results can be found in Table 4, MLP-ONLY consists of only the MLP module, MLP-F.F. adopts the fixed-frequency strategy during training, and MLP-COPY represents the model that incorporates the copy mechanism. It can be observed that MLP-F.F. outperforms MLP-COPY significantly, while MLP-COPY performs even slightly worse than the original MLP-ONLY. This suggests that MLP-COPY fails to effectively incorporate historical frequency information during the training process. Furthermore, by considering Figure 5, we can observe that MLP-COPY not only fails to enhance the model performance, but also significantly prolongs the training time for each epoch. This indirectly confirms the superiority of the fixed-frequency strategy adopted by SiMFy.

## 6.5 Discussion on the Depth of MLP

SiMFy uses a simple one-layer MLP to draw the final prediction. We also want to figure out whether the performance of the model will improve or worsen when the one-layer MLP is replaced with some deeper or more complicated neural structure. To answer this question, we conduct experiments to replace the one-layer MLP with deeper ones. The experimental results on the ICEWS14 dataset are presented in Table 5. From this table, we can see that as the structure of the model becomes more complex, its performance has hardly changed. This is because the complex model structure, while increasing computational costs, does not better capture the evolving information of entities and relations.

Table 5: Comparative experiments on the depth of MLP on ICEWS14

| Model | MRR | Hits@1 | Hits@3 | Hits@10 |
|---|---|---|---|---|
| **SiMFy-1MLP** | **39.54** | 29.56 | **44.56** | **59.18** |
| SiMFy-2MLP | 39.49 | **29.74** | 43.97 | 58.58 |
| SiMFy-3MLP | 39.29 | **29.74** | 43.46 | 58.20 |
| SiMFy-RNN+MLP | 39.27 | 29.45 | 43.89 | 58.36 |

## 7 Conclusion

In this paper, we have proposed a new model, called SiMFy, for the TKG reasoning task. SiMFy is a conceptually straightforward method that simply combines MLP and historical frequency to model the temporal events in the TKGs. The experimental results demonstrate that SiMFy not only outperforms many existing complex methods but also exhibits faster convergence speed and better generalization ability. Our findings suggest that a well-designed MLP-based model, such as SiMFy, can effectively address the limitations faced by complex architectures, making it a practical and efficient solution for the TKG reasoning task.

## Limitations

One limitation of this paper is that no in-depth exploration of the potential mutual influence between historical events and unseen events is considered, which means that events that have been repeatedly happening in the sequence of historical KGs may not happen in the future, and instead, new unseen events will take place. Another limitation lies in that the time span for calculating the frequency of historical events is too long, and a more precise time window is needed to better capture the long- and short-term evolutionary patterns of events.

## Acknowledgements

This work was supported in part by the National Key R&D Program of China under Grant 2020AAA0108501, and the Key Program of Hubei under Grant JD2023008.

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

# A Appendix

## A.1 Results with Filtered Metrics

Table 6 provides the experimental results (with filtered metrics) of SiMFy compared with other baselines on ICEWS14, ICEWS18, ICEWS05-15, and GDELT datasets.

## A.2 Statistics of Datasets

The detailed statistics of the ICEWS14, ICEWS18, ICEWS05-15, and GDELT datasets are presented in Table 7.

## A.3 Proportion of Unseen Events

We conduct a statistical analysis on the proportion of unseen events in the test set of the ICEWS14, ICEWS18, ICEWS05-15, and GDELT datasets. The results are presented in Table 8.

## A.4 Supplementary Figures on Convergence Speed and Generalization

We provide the missing figures on the ICEWS14 and ICEWS18 datasets to supplement the Section 6.1. The results on ICEWS14 are shown in Figure 6 and the results on ICEWS18 are shown in Figure 7.

Table 6: Experimental results of the entity prediction task (**filtered metrics**) on ICEWS14, ICEWS18, ICEWS05-15, and GDELT datasets. The best results are boldfaced and the ones of the second-best ones are underlined.

| Model | ICEWS14 (filtered) | | | | ICEWS18 (filtered) | | | | ICEWS05-15 (filtered) | | | | GDELT (filtered) | | | |
|---|---|---|---|---|---|---|---|---|---|---|---|---|---|---|---|---|
| | MRR | Hits@1 | Hits@3 | Hits@10 | MRR | Hits@1 | Hits@3 | Hits@10 | MRR | Hits@1 | Hits@3 | Hits@10 | MRR | Hits@1 | Hits@3 | Hits@10 |
| xERTE | 40.79 | 32.70 | 45.67 | 57.30 | 29.31 | 21.03 | 33.51 | 46.48 | 46.62 | 37.84 | 52.31 | 63.92 | - | - | - | - |
| RE-NET | 45.71 | 38.42 | 49.06 | 59.12 | 42.93 | 36.19 | 45.47 | 55.80 | 42.97 | 31.26 | 46.85 | 63.47 | 40.12 | 32.43 | 43.40 | 53.80 |
| CyGNet | 48.63 | 41.77 | 52.50 | 60.29 | 46.69 | _40.58_ | 49.82 | 57.14 | 57.00 | 50.06 | 61.51 | 68.47 | **51.56** | **45.63** | **54.86** | **60.96** |
| CENET | _52.87_ | _47.94_ | _54.42_ | _62.79_ | **50.09** | **46.02** | **51.05** | _57.85_ | - | - | - | - | - | - | - | - |
| CEN | 37.34 | 28.04 | 41.09 | 55.76 | 28.21 | 19.10 | 31.80 | 45.89 | 37.53 | 27.57 | 41.49 | 57.51 | 19.71 | 12.47 | 20.98 | 33.82 |
| RE-GCN | 38.27 | 28.40 | 42.54 | 57.88 | 29.21 | 19.79 | 32.77 | 47.54 | 38.68 | 28.26 | 43.44 | 59.11 | 29.46 | 21.74 | 32.01 | 43.62 |
| **SiMFy** | **54.81** | **47.99** | **58.54** | **66.65** | _46.87_ | 39.29 | _51.00_ | 60.23 | **60.76** | **53.43** | **65.62** | **72.94** | _47.40_ | _40.17_ | _50.81_ | _60.46_ |

Table 7: Statistics of the datasets

| Dataset | Entities | Relations | Train | Valid | Test | Timestamp | Granularity |
|---|---|---|---|---|---|---|---|
| ICEWS14 | 7128 | 230 | 74845 | 8514 | 7371 | 365 | 24 hours |
| ICEWS18 | 23033 | 256 | 373018 | 45995 | 49545 | 304 | 24 hours |
| ICEWS05-15 | 10488 | 251 | 368868 | 46302 | 46159 | 4017 | 24 hours |
| GDELT | 7691 | 240 | 1734399 | 238765 | 305241 | 2751 | 15 minutes |

Table 8: The proportion of unseen events

| Dataset | Test | Unseen Events | Repetitive Events | Proportion of Unseen Events |
|---|---|---|---|---|
| ICEWS14 | 7371 | 3511 | 3860 | 47% |
| ICEWS18 | 49545 | 24560 | 24985 | 49% |
| ICEWS05-15 | 46159 | 14592 | 31567 | 31% |
| GDELT | 305241 | 107062 | 198179 | 35% |

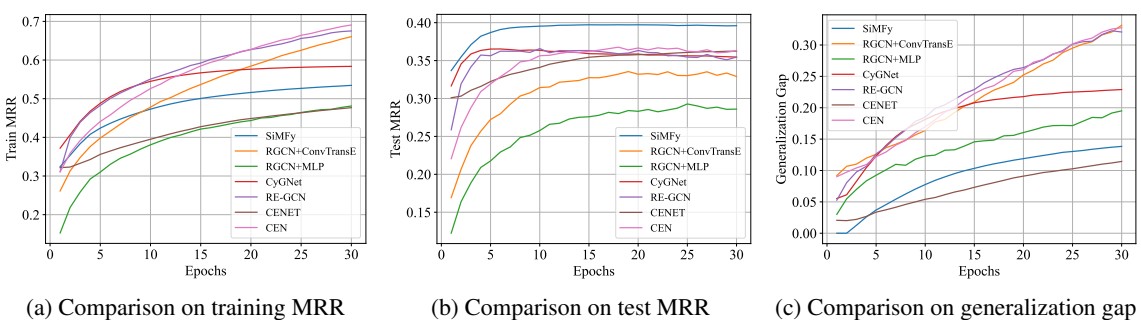

(a) Comparison on training MRR    (b) Comparison on test MRR    (c) Comparison on generalization gap

Figure 6: Comparison on the training MRR, test MRR, and generalization gap on ICEWS14

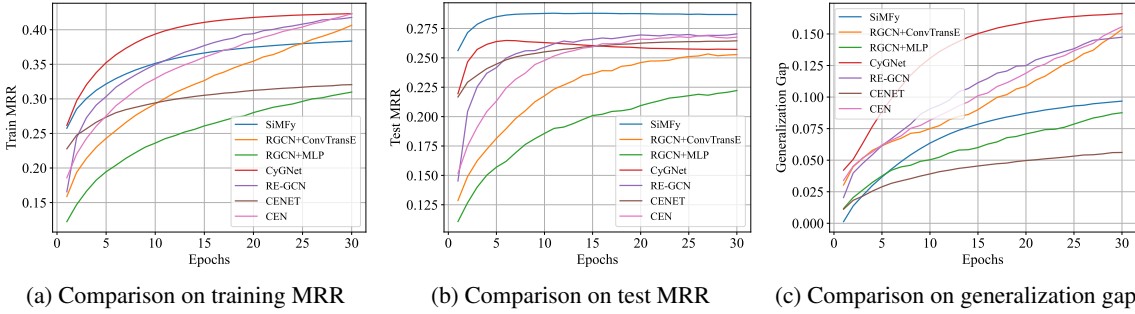

(a) Comparison on training MRR    (b) Comparison on test MRR    (c) Comparison on generalization gap

Figure 7: Comparison on the training MRR, test MRR, and generalization gap on ICEWS18