# OpenReview forum: "SiMFy: A Simple Yet Effective Approach for Temporal Knowledge Graph Reasoning"
_EMNLP/2023/Conference — EMNLP 2023 Findings_

### Official Review · Reviewer_AXyf · 2023-07-31

**Soundness:** 4

**Excitement:**

4: Strong: This paper deepens the understanding of some phenomenon or lowers the barriers to an existing research direction.

**Paper Topic And Main Contributions:**

The paper explores the effectiveness of complex model architectures in Temporal Knowledge Graph (TKG) reasoning, a critical component of knowledge graph completion. The authors highlight issues with the current models, such as low training efficiency and inadequate generalization ability. To address these concerns, they propose a simpler approach named SiMFy, which employs a multilayer perceptron (MLP) to model the structural dependencies of events and incorporates historical frequency during inference using a fixed-frequency strategy. The significant contributions of this research are: 1) Demonstrating that SiMFy can achieve state-of-the-art performance, outperforming complex models in faster convergence speed and better generalization ability; 2) Showing that SiMFy consumes less time in the training process; and 3) Providing evidence that SiMFy is better able to capture the structural dependencies of events in KGs. The results suggest that replacing complex models with simpler counterparts is feasible.

**Questions For The Authors:**

1. The authors use Historical Frequency Learning to contain temporal features. However, there are many ways to deal with the temporal features. Why did the authors finally pick this frequency-based method? What about other similar temporal methods?
2. The proposed method is intrinsically similar to a self-attention mechanism, as they all use attention-based scores to leverage the sequential encodings. More discussion on the main difference and connection with the attention-based models is recommended.
3. What if the timestamp t becomes significantly large? Could there be forgetting issues like an RNN model?
4. The authors use a simple MLP to draw the final prediction. What if the MLP is replaced with some deeper or more complicated neural structure? Will the performance become better or worse, and why?
5. More discussion on the motivation of the proposed method is recommended.

**Reasons To Accept:**

1. The paper is well-written and easy to follow.
2. The paper might potentially impact the understanding of the knowledge graph.
3. The empirical results are promising and impressive.
4. The proposed methods are simple but efficient, and the frequency-based mechanism is novel in the NLP community.

**Reasons To Reject:**

1. The part related to Russia and Ukraine needs an academic reference. The authors should know that an academic paper must be rigorous: any information presented should be clear and objective, properly citing academic references. An academic paper should be responsible for elucidating academic facts and should not be a medium of historical and political events.
2. The motivation and theoretical validation are insufficient to demonstrate the superiority of the proposed method.

**Reproducibility:**

4: Could mostly reproduce the results, but there may be some variation because of sample variance or minor variations in their interpretation of the protocol or method.

**Reviewer Confidence:**

3: Pretty sure, but there's a chance I missed something. Although I have a good feel for this area in general, I did not carefully check the paper's details, e.g., the math, experimental design, or novelty.

---

> ### Author Rebuttal · Authors · 2023-08-28
>
> First of all, thank the reviewer for pointing out the issue regarding the example involving Russia and Ukraine. The concern is valid, and we will either add the correct reference or choose a more suitable example to explain the TKG reasoning task.
>
> **Q1: More discussion on the motivation of the proposed method is recommended.**
>
> **A1:** As required by the newly added track "Efficient Methods for NLP" at this year's conference, the direct motivation of SiMFy is to achieve comparable effectiveness when compared with complex models, but in a simpler and more efficient manner. We have found that through our designed approach, we can achieve comparable performance to existing models while having lower training costs and better generalization ability. As agreed by **Reviewer G2Sv**, we believe our work could not only shed some light on the future algorithm design for TKG reasoning but also inspire future studies to rethink the importance of simple model architectures rather than a complicated one.
>
> **Q2: The authors use Historical Frequency Learning to contain temporal features. However, there are many ways to deal with the temporal features. Why did the authors finally pick this frequency-based method? What about other similar temporal methods?**
>
> **A2:** In fact, we have provided detailed explanations and justifications for the selection of the Historical Frequency Learning strategy in **Section 6.4**. We have indeed tried many other methods to model temporal features, such as the time-encoding function used in GraphMixer [1] and traditional widely-used approaches like utilizing RNN to contain temporal features during the training process. However, through empirical study, we found that the strategy adopted by SiMFy not only achieves optimal performance but also has higher efficiency and generalization. Therefore, we choose Historical Frequency Learning to contain temporal features in our model.
>
> **Q3: The proposed method is intrinsically similar to a self-attention mechanism, as they all use attention-based scores to leverage the sequential encodings. More discussion on the main difference and connection with the attention-based models is recommended.**
>
> **A3:** Although it may seem similar at first glance, we must point out that there are significant differences between SiMFy's modeling approach for temporal quadruple sequences and the self-attention mechanism. In terms of self-attention mechanism, weights should be calculated for each quadruple **(s, r, o, t)** in different timestamps to represent the correlation between this quadruple and other quadruples in the sequence. However, SiMFy separates the entity-relation triple **(s, r, o)** from the corresponding quadruple and calculates the similarity scores between specific query **(s, r, ?)** and all entities first. As for the temporal dimension **t**, it is processed separately using the Historical Frequency Learning strategy based on chronological order.
>
> **Q4: What if the timestamp t becomes significantly large? Could there be forgetting issues like an RNN model?**
>
> **A4:** In general, forgetting issues definitely exist. However, in the scenario of TKG reasoning, we actually need the model to appropriately "forget" some previous sequence information sometimes. This is because in the TKG, as time goes by, the connections between entities and relations in the KG are constantly evolving. For the TKG reasoning task, as clarified by CEN [2], we are more concerned about the recent evolutional pattern of KG sequence.
>
> **Q5: The authors use a simple MLP to draw the final prediction. What if the MLP is replaced with some deeper or more complicated neural structure? Will the performance become better or worse, and why?**
>
> **A5:** To answer this question, here we add experiments to replace the MLP with some deeper and more complicated neural structures, e.g., multi-layer MLP and RNN. The experimental results on the ICEWS14 dataset are presented as follows:
>
> | ICEWS14         |         |         |         |         |
> |:---------------:|:-------:|:-------:|:-------:|:-------:|
> |                 | **MRR**     | **Hits@1**  | **Hits@3**  | **Hits@10** |
> | **SiMfy\(1MLP\)**     | **39\.54**  | 29\.56  | **44\.56**  | **59\.18**  |
> | SiMfy\(2MLP\)     | 39\.49  | 29\.74  | 43\.97  | 58\.58  |
> | SiMfy\(3MLP\)     | 39\.29  | 29\.74  | 43\.46  | 58\.20  |
> | SiMfy\(RNN\+MLP\) | 39\.27  | 29\.45  | 43\.89  | 58\.36  |
>
> From this table, we can see that as the structure of the model becomes more complex, its performance has even slightly worsened. We think that this is because the complex model structure, while increasing computational costs, does not better capture the evolving information of entities and relations. However, more specific reasons would require further and extensive experimentation to investigate, which are left to our future work.
>
> [1] Weilin Cong, Si Zhang, Jian Kang, Baichuan Yuan, Hao Wu, Xin Zhou, Hanghang Tong, and Mehrdad Mahdavi. 2023. Do we really need complicated model architectures for temporal networks? arXiv preprint arXiv:2302.11636.
> [2] Zixuan Li, Saiping Guan, Xiaolong Jin, Weihua Peng, Yajuan Lyu, Yong Zhu, Long Bai, Wei Li, Jiafeng Guo, and Xueqi Cheng. 2022. Complex evolutional pattern learning for temporal knowledge graph reasoning. In Proceedings of the 60th Annual Meeting of the Association for Computational Linguistics.

---

### Official Review · Reviewer_Tw4C · 2023-08-04

**Soundness:** 3

**Excitement:**

3: Ambivalent: It has merits (e.g., it reports state-of-the-art results, the idea is nice), but there are key weaknesses (e.g., it describes incremental work), and it can significantly benefit from another round of revision. However, I won't object to accepting it if my co-reviewers champion it.

**Paper Topic And Main Contributions:**

This paper presents a simple approach for temporal KG reasoning. It uses just an MLP instead of more sophisticated GNN-based architectures proposed in the literature. This leads to training efficiency and better generalization performance. The ablation studies provide evidence for the design.

**Questions For The Authors:**

1. Could you also provide the filtered setting metrics for a fair comparison?

**Reasons To Accept:**

1. They propose a simple model architecture that works better
2. It leads to training efficiency and better generalization performance.
3. It has better convergence than other models.

**Reasons To Reject:**

1. The authors report results on raw Hits@k and MRR. Whereas all the well-cited works report results on filtered metrics. Though the authors argue that the filtered setting has little practical relevance, all of the prior works (CyGNet, CENET, ) use that setting and not the raw one. The authors should have reported both metrics for a fair comparison.
\textbf{Update}: The authors agreed to provide results in the time-aware filtered setting which is widely used in literature and is also a fair metric.
2. The choice of datasets is suspect. The GDELT [1] dataset is missing but has been used in all strong baselines.

[1] Sun, Haohai, et al. "Graph Hawkes Transformer for Extrapolated Reasoning on Temporal Knowledge Graphs." Proceedings of the 2022 Conference on Empirical Methods in Natural Language Processing. 2022.

**Reproducibility:**

4: Could mostly reproduce the results, but there may be some variation because of sample variance or minor variations in their interpretation of the protocol or method.

**Reviewer Confidence:**

4: Quite sure. I tried to check the important points carefully. It's unlikely, though conceivable, that I missed something that should affect my ratings.

---

> ### Author Rebuttal · Authors · 2023-08-28
>
> We first thank the reviewer for the constructive comments and suggestions.
>
> **Q1: The reviewer is wondering whether the authors run experiments on the larger dataset GDELT.**
>
> **A1:** Taking this comment into consideration, we conduct experiments on the larger dataset GDELT (under raw setting, i.e., using the original ranking list predicted by the model). The results are shown in the table below:
>
> | **GDELT**   |         |         |         |         |
> |:-------:|:-------:|:-------:|:-------:|:-------:|
> |         | **MRR**     | **Hits@1**  | **Hits@3**  | **Hits@10** |
> | xERTE   | \-      | \-      | \-      | \-      |
> | RE\-NET | 19\.60  | 12\.03  | 20\.56  | 33\.89  |
> | CyGNet  | 17\.99  | 11\.10  | 19\.03  | 31\.26  |
> | CENET   | \-      | \-      | \-      | \-      |
> | GHT     | 20\.04  | 12\.68  | 21\.37  | 34\.42  |
> | CEN     | 19\.27  | 11\.96  | 20\.60  | 33\.50  |
> | RE\-GCN | 19\.07  | 12\.16  | 20\.20  | 32\.39  |
> | **SiMFy**   | **21\.33**  | **13\.38**  | **23\.16**  | **36\.87**  |
>
> It can be seen that, similar to the results on the ICEWS datasets, SiMFy outperforms all other baselines on the GDELT dataset. It should be noted that when we ran the source code provided by xERTE [1] and CENET [2] on GDELT, the running programs of xERTE and CENET  crashed due to the out-of-memory issue.
>
> **Q2: Could you also provide the filtered setting metrics for a fair comparison?**
>
> **A2:** First and foremost, it is important for us to explain why we choose the raw setting instead of the filtered setting in the scenario of TKG reasoning.
>
> In fact, the filtered setting proposed by TransE[3] has been widely used in tasks related to **common knowledge graph completion**. However, in the scenario of TKG reasoning, the filtered setting is no longer applicable due to the introduction of timestamp **t**. Let's take a simple example: given a test quadruple **(Donald Trump, visit, Japan, 2019-05-25)**, and we perform the object prediction **(Donald Trump, visit, ?, 2019-05-25)**. We have observed the quadruple **(Donald Trump, visit, England, 2018-07-13)** in the training set. According to the filtered setting, **(Donald Trump, visit, England)** will be considered as a genuine triple at the timestamp **2019-05-25** and will be wrongly predicted as the correct answer because the triple **(Donald Trump, visit, England)** appears in the training set in the quadruple **(Donald Trump, visit, England, 2018-07-13)**. However, the triple **(Donald Trump, visit, England)** is only temporally valid on **2018-07-13** but not on **2019-05-25**. Thus, the filtered setting may probably get incorrect higher ranking scores. This also explains why **xERTE** [1], **CEN** [4], **RE-GCN** [5], and **GHT** [6] did not adopt the filtered setting.
>
> Here we also provide the experimental results under the filtered setting as follows:
>
> | GDELT\(filtered\) |         |         |         |         | ICEWS14\(filtered\) |         |         |         |         |
> |:-----------------:|:-------:|:-------:|:-------:|:-------:|:-------------------:|:-------:|:-------:|:-------:|:-------:|
> |                   | **MRR**     | **Hits@1**  | **Hits@3**  | **Hits@10** |                     | **MRR**     | **Hits@1**  | **Hits@3**  | **Hits@10** |
> | xERTE             | crashed |         |         |         | xERTE               | 40\.79  | 32\.70  | 45\.67  | 57\.30  |
> | RE\-NET           | 40\.12  | 32\.43  | 43\.40  | 53\.80  | RE\-NET             | 45\.71  | 38\.42  | 49\.06  | 59\.12  |
> | CyGNet            | **51\.56**  | **45\.63**  | **54\.86**  | **60\.96**  | CyGNet              | 48\.63  | 41\.77  | 52\.50  | 60\.29  |
> | CENET             | crashed |         |         |         | CENET               | 52\.87  | 47\.94  | 54\.42  | 62\.79  |
> | GHT               | \-      | \-      | \-      | \-      | GHT                 | \-      | \-      | \-      | \-      |
> | CEN               | 19\.71  | 12\.47  | 20\.98  | 33\.82  | CEN                 | 37\.34  | 28\.04  | 41\.09  | 55\.76  |
> | RE\-GCN           | 29\.46  | 21\.74  | 32\.01  | 43\.62  | RE\-GCN             | 38\.27  | 28\.40  | 42\.54  | 57\.88  |
> | SiMFy             | 47\.40  | 40\.17  | 50\.81  | 60\.46  | SiMFy              | **54\.81**  | **47\.99**  | **58\.54**  | **66\.65**  |
>
> | ICEWS18\(filtered\) |         |         |         |         | ICEWS05\-15\(filtered\) |         |         |         |         |
> |:-------------------:|:-------:|:-------:|:-------:|:-------:|:-----------------------:|:-------:|:-------:|:-------:|:-------:|
> |                     | **MRR**     | **Hits@1**  | **Hits@3**  | **Hits@10** |                         | **MRR**     | **Hits@1**  | **Hits@3**  | **Hits@10** |
> | xERTE               | 29\.31  | 21\.03  | 33\.51  | 46\.48  | xERTE                   | 46\.62  | 37\.84  | 52\.31  | 63\.92  |
> | RE\-NET             | 42\.93  | 36\.19  | 45\.47  | 55\.80  | RE\-NET                 | 42\.97  | 31\.26  | 46\.85  | 63\.47  |
> | CyGNet              | 46\.69  | 40\.58  | 49\.82  | 57\.14  | CyGNet                  | 57\.00  | 50\.06  | 61\.51  | 68\.47  |
> | CENET               | **50\.09**  | **46\.02**  | **51\.05**  | 57\.85  | CENET                   | crashed |         |         |         |
> | GHT                 | \-      | \-      | \-      | \-      | GHT                     | \-      | \-      | \-      | \-      |
> | CEN                 | 28\.21  | 19\.10  | 31\.80  | 45\.89  | CEN                     | 37\.53  | 27\.57  | 41\.49  | 57\.51  |
> | RE\-GCN             | 29\.21  | 19\.79  | 32\.77  | 47\.54  | RE\-GCN                 | 38\.68  | 28\.26  | 43\.44  | 59\.11  |
> | SiMFy               | 46\.87  | 39\.29  | 51\.00  | **60\.23**  | SiMFy                   | **60\.76**  | **53\.43**  | **65\.62**  | **72\.94**  |
>
> It needs to be clarified that **GHT** [6] used a raw setting in their paper, but the authors provide neither the filtered-version results nor the implemented source code. "**Crashed**" denotes that their program crashed when running on a specific dataset. From this table, we can observe that, under the filtered setting, all models obtain incorrect higher ranking scores compared to their performance under the raw setting. Although SiMFy still performs the best on the ICEWS14 and ICEWS05-15 datasets, it  performs second-best on the GDELT and ICEWS18 datasets. We argue that these experimental results obtained under the incorrect evaluation criteria (filtered setting) have little practical significance.
>
> [1] Zhen Han, Peng Chen, Yunpu Ma, and Volker Tresp. 2021. Explainable subgraph reasoning for forecasting on temporal knowledge graphs. In International Conference on Learning Representations.
> [2] Yi Xu, Junjie Ou, Hui Xu, and Luoyi Fu. 2022. Temporal knowledge graph reasoning with historical contrastive learning. arXiv preprint arXiv:2211.10904
> [3] Bordes A, Usunier N, Garcia-Duran A, et al. Translating embeddings for modeling multi-relational data[J]. Advances in neural information processing systems, 2013, 26.
> [4] Zixuan Li, Saiping Guan, Xiaolong Jin, Weihua Peng, Yajuan Lyu, Yong Zhu, Long Bai, Wei Li, Jiafeng Guo, and Xueqi Cheng. 2022. Complex evolutional pattern learning for temporal knowledge graph reasoning. In Proceedings of the 60th Annual Meeting of the Association for Computational Linguistics.
> [5] Zixuan Li, Xiaolong Jin, Wei Li, Saiping Guan, Jiafeng Guo, Huawei Shen, Yuanzhuo Wang, and Xueqi Cheng. 2021b. Temporal knowledge graph reasoning based on evolutional representation learning. In Proceedings of the 44th International ACM SIGIR Conference on Research and Development in Information Retrieval, pages 408–417.
> [6] Sun, Haohai, et al. "Graph Hawkes Transformer for Extrapolated Reasoning on Temporal Knowledge Graphs." Proceedings of the 2022 Conference on Empirical Methods in Natural Language Processing. 2022.

---

### Official Review · Reviewer_G2Sv · 2023-08-04

**Typos Grammar Style And Presentation Improvements:** N.A.
**Soundness:** 3

**Excitement:**

4: Strong: This paper deepens the understanding of some phenomenon or lowers the barriers to an existing research direction.

**Justification For Ethical Concerns:**

N.A.

**Missing References:**

Rakshit Trivedi, Hanjun Dai, Yichen Wang, and Le Song. Know-evolve: Deep temporal reasoning for dynamic knowledge graphs. In Proceedings of the 34th International Conference on Machine Learning, volume 70, pages 3462–3471. JMLR. org, 2017.

Zhen Han, Zifeng Ding, Yunpu Ma, Yujia Gu, and Volker Tresp. 2021. Learning Neural Ordinary Equations for Forecasting Future Links on Temporal Knowledge Graphs. In Proceedings of the 2021 Conference on Empirical Methods in Natural Language Processing, pages 8352–8364, Online and Punta Cana, Dominican Republic. Association for Computational Linguistics.

**Paper Topic And Main Contributions:**

The paper studied the topic of temporal knowledge graph reasoning, especially on forecasting future event based on the information of structural dependency, temporality of entities and relations, as well as historical event patterns. The paper argues that the existing approaches are conceptually and technically complex, and are even over-designed regarding model architectures such that they have insufficient generalization ability. The authors proposed a simple yet effective approach, i.e., use a one-layer MLP to jointly encode entities and relations to capture structural and temporal information and get a preliminary candidate entity score, then calculate the historical frequency score of candidate entities. The final entity candidate score is a weighted average of the above two scores. The author empirically show that this simple approach has higher training efficiency and better generalization ability while can achieving comparable performance comparable performance to SoTA approaches with more complex architectures. Besides, they found MLP performs comparably to GNN in capturing structural dependency information for the TKG reasoning task. Additionally, the author demonstrates the effectiveness of using simple features to model the repetitive pattern of TKGs instead of directly incorporate historical frequency information of entities during training.


**Questions For The Authors:**

1. The reviewer is wondering whether the authors run experiments on larger dataset such GDELT dataset, which contains several million quadruples.

2. As author stated in Section 6.1, the GCN-based models have overfitting problems. How are the model parameter numbers of GCN-based methods compared to SiMFy? Does the author try to reduce the model parameters of GCN-based method and see better performance?


**Reasons To Accept:**

The paper shows a simple approach combining one-layer MLP encoding and historical event frequency information achieved comparable or even better results than existing SoTA models on ICEWS datasets. The paper also demonstrates that the proposed approach has a faster convergence speed than existing SoTA methods and stronger generalization ability than CyGNet, RE-GCN etc. The author reports many thorough and interesting in-depth analysis regarding the effectiveness between GNN and MLP on tKG reasoning, which would benefit future research.


**Reasons To Reject:**

The paper argues a simple MLP-based approach could beat SoTA models with complex model architecture. However, the paper only report the results on ICEWS datasets (14, 18, 05-15) but only on other tKG datasets such as GDELT, which is larger and more complex. The reviewer concerns that the observations on large-scale datasets maybe different.


**Reproducibility:**

5: Could easily reproduce the results.

**Reviewer Confidence:**

5: Positive that my evaluation is correct. I read the paper very carefully and I am very familiar with related work.

---

> ### Author Rebuttal · Authors · 2023-08-28
>
> We thank the reviewer for the constructive comments and suggestions.
>
> **Q1: The reviewer is wondering whether the authors run experiments on larger dataset such as the GDELT dataset, which contains several million quadruples.**
>
> **A1:** Based on this comment, we did run experiments on the large dataset GDELT, and the results are shown in the following table:
>
> | **GDELT**   |         |         |         |         |
> |:-------:|:-------:|:-------:|:-------:|:-------:|
> |         | **MRR**     | **Hits@1**  | **Hits@3**  | **Hits@10** |
> | xERTE   | \-      | \-      | \-      | \-      |
> | RE\-NET | 19\.60  | 12\.03  | 20\.56  | 33\.89  |
> | CyGNet  | 17\.99  | 11\.10  | 19\.03  | 31\.26  |
> | CENET   | \-      | \-      | \-      | \-      |
> | GHT     | 20\.04  | 12\.68  | 21\.37  | 34\.42  |
> | CEN     | 19\.27  | 11\.96  | 20\.60  | 33\.50  |
> | RE\-GCN | 19\.07  | 12\.16  | 20\.20  | 32\.39  |
> | **SiMFy**   | **21\.33**  | **13\.38**  | **23\.16**  | **36\.87**  |
>
> It can be seen that, similar to the results on the ICEWS datasets, our SiMFy consistently outperforms all other baselines on the GDELT dataset. It should be noted that when we ran the source code provided by xERTE [1] and CENET [2] on GDELT, the running programs of xERTE and CENET crashed due to the out-of-memory issue.
>
> **Q2: As author stated in Section 6.1, the GCN-based models have overfitting problems. How are the model parameter numbers of GCN-based methods compared to SiMFy? Does the author try to reduce the model parameters of GCN-based method and see better performance?**
>
> **A2:** The model parameters of SiMFy are much smaller than those of GCN-based methods. This is because our SiMFy is simply composed of an MLP layer, which maps the input embedding to the output embedding. In addition to these options, the GCN-based models also perform the multi-step propagation operation, which will significantly increase the model parameters, especially when the depth of the GNN is increased.
>
> Here, we also vary the model parameters of GCN-based models to investigate its impact on the performance. We select two representative GCN-based models, i.e., CEN [3] and RGCN+ConvTransE, as target, and conduct experiments on the ICEWS14 dataset. Specifically, we reduce their model parameters via decreasing the depth (number of layers) of GNNs from 2 to 1, and decreasing the hidden units of each layer from 200 to 100. The experimental results are shown as below:
>
> | CEN     |              |         |         |         |         | RGCN\+ConvTransE |              |         |         |         |         |
> |:-------:|:------------:|:-------:|:-------:|:-------:|:-------:|:----------------:|:------------:|:-------:|:-------:|:-------:|:-------:|
> | layers  | hidden units | **MRR**     | **Hits@1**  | **Hits@3**  | **Hits@10** | layers           | hidden units | **MRR**     | **Hits@1**  | **Hits@3**  | **Hits@10** |
> | 2\(\*\) | 200\(\*\)    | **36\.32**  | **26\.83**  | 40\.20  | 54\.93  | 2\(\*\)          | 200\(\*\)    | **35\.07**  | **26\.17**  | **39\.15**  | **52\.23**  |
> | 2       | 100          | 36\.06  | 26\.29  | **40\.38**  | **55\.15**  | 2                | 100          | 33\.98  | 24\.70  | 38\.04  | 52\.06  |
> | 1       | 200          | 35\.93  | 26\.61  | 39\.81  | 54\.46  | 1                | 200          | 33\.53  | 24\.39  | 37\.14  | 52\.12  |
> | 1       | 100          | 35\.99  | 26\.38  | 40\.29  | 54\.42  | 1                | 100          | 33\.89  | 24\.49  | 38\.25  | 51\.66  |
>
> In this table, the parameters marked with "*" are the default settings (i.e., 2 layers and 200 hidden units) for the model. It can be observed that when reducing the parameters of GCN-based models, the performance of the models will accordingly decrease to varying degrees. This demonstrates that the performance of GCN-based models is closely related to the number of model parameters. This is also the reason why GCN-based models are prone to be overfitted.
>
> [1] Zhen Han, Peng Chen, Yunpu Ma, and Volker Tresp. 2021. Explainable subgraph reasoning for forecasting on temporal knowledge graphs. In International Conference on Learning Representations.
> [2] Yi Xu, Junjie Ou, Hui Xu, and Luoyi Fu. 2022. Temporal knowledge graph reasoning with historical contrastive learning. arXiv preprint arXiv:2211.10904
> [3] Zixuan Li, Saiping Guan, Xiaolong Jin, Weihua Peng, Yajuan Lyu, Yong Zhu, Long Bai, Wei Li, Jiafeng Guo, and Xueqi Cheng. 2022. Complex evolutional pattern learning for temporal knowledge graph reasoning. In Proceedings of the 60th Annual Meeting of the Association for Computational Linguistics.

---

### Meta-Review · Area_Chair_n9ud · 2023-09-15

**Recommendation:** 4

**Metareview:**

This paper evaluates the necessity of complex models for temporal knowledge graph reasoning and introduces a streamlined approach, SiMFy, that uses a multilayer perceptron (MLP) for modeling event structural dependencies and employs a fixed-frequency strategy for inference. SiMFy achieves comparable performance to SoTA approaches with faster convergence, reduced training time, and capturing structural dependency information for the TKG reasoning task.

This paper is technically sound, and the experiments are conducted appropriately, although the impact of the results is relatively incremental. As reviewers point out, experiments on larger datasets such as the GDELT dataset, will strengthen the paper. More importantly, as reviewer AXyf pointed out, the example of the Russia-Ukraine conflict is inappropriate from an academic perspective. Additionally, using TKG for predicting the future is less sensible. It would be more useful and appealing to identify similar event structures from the past for reference.

---

### Decision · Program_Chairs · 2023-10-07

**Decision:**

Accept-Findings

**Comment:**

This paper evaluates the necessity of complex models for temporal knowledge graph reasoning and introduces a streamlined approach, SiMFy, that uses a multilayer perceptron (MLP) for modeling event structural dependencies and employs a fixed-frequency strategy for inference. SiMFy achieves comparable performance to SoTA approaches with faster convergence, reduced training time, and capturing structural dependency information for the TKG reasoning task.

This paper is technically sound, and the experiments are conducted appropriately, although the impact of the results is relatively incremental. As reviewers point out, experiments on larger datasets such as the GDELT dataset, will strengthen the paper. More importantly, as reviewer AXyf pointed out, the example of the Russia-Ukraine conflict is inappropriate from an academic perspective. Additionally, using TKG for predicting the future is less sensible. It would be more useful and appealing to identify similar event structures from the past for reference.